
# Pseudogap metal and magnetization plateau from doping moiré Mott insulator

**Yang Zhang⋆ and Liang Fu**

Department of Physics, Massachusetts Institute of Technology, Cambridge, MA 02139, USA

## Abstract

The problem of doping Mott insulators is of fundamental importance and long-standing interest in the study of strongly correlated electron systems. The advent of semiconductor based moiré materials opens a new ground for simulating the Hubbard model on the triangular lattice and exploring its rich phase diagram as a function of doping and external magnetic field. Based on our recent identification of spin polaron quasiparticle in Mott insulator [1], in this work we predict the emergence of a pseudogap metal phase at small doping below half filling and an intermediate range of fields, which exhibits a single-particle gap and a doping-dependent magnetization plateau.

---

As the paradigmatic model for strongly correlated electron systems, the Hubbard model captures in the simplest form the essence of numerous electronic phenomena [2], including metal-insulator transition, metallic ferromagnetism, charge/spin stripe states and unconventional superconductivity. The Hubbard model was studied intensively in the context of cuprate high temperature superconductors [3]. Recently, transition metal dichalcogenide (TMD) moiré heterostructures have emerged as a robust and tunable platform for the realization of Hubbard model physics [4–9]. The moiré superlattice due to lattice mismatch or rotational twist introduces a long-wavelength periodic potential for itinerant charge carriers. At large moiré period, the system is akin to an array of artificial atoms, where each "atom" corresponds to a local minimum of the moiré potential and neighboring atoms are weakly coupled by tunneling through the potential barrier. As a result, narrow moiré bands are formed, and the band dispersion is well described by a tight binding model on the triangular or honeycomb lattice [4,5]. By further including Coulomb repulsion, a Hubbard model with on-site and non-local interactions [10] is obtained as the effective Hamiltonian of TMD moiré superlattices.

Compared to other Hubbard model materials, TMD moiré superlattice has a distinct advantage due to its robustness and tunability. The formation of narrow moiré bands needed for Mott-Hubbard physics does not require a magic twist angle. The band filling can be tuned continuously by electrostatic gating. The non-local interaction in moiré Hubbard systems can be screened by metallic gates [11]. The moiré bandwidth can be tuned by the twist angle and out-of-plane electric field [8,12].

---

⋆ Current address: Department of Physics and Astronomy, University of Tennessee, Knoxville, Tennessee 37996, USA; Min H. Kao Department of Electrical Engineering and Computer Science, University of Tennessee, Knoxville, Tennessee 37996, USA.

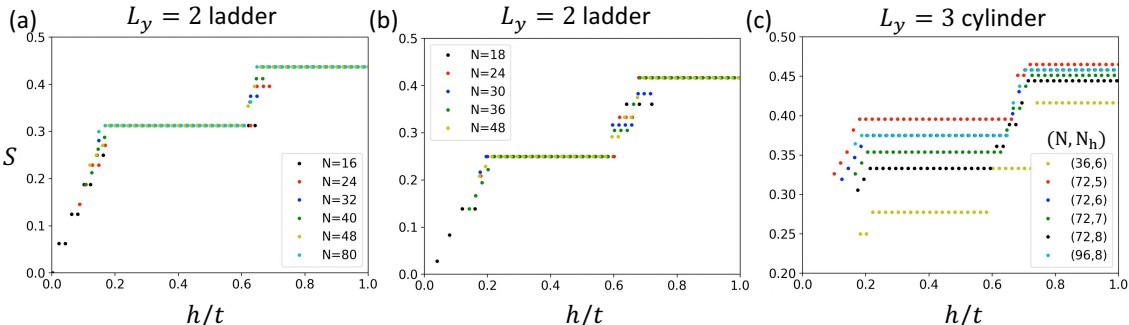

Figure 1: The total spin $S$ under magnetic field obtained by DMRG at infinite U limit. Finite-size scaling of (a) $\delta = 1/8$ and (b) $\delta = 1/6$ hole doping on two-leg ladder with $L_x$ up to 80. (c) Doping dependent magnetization plateaus in $L_x = 24, 32$ three-leg cylinders.

As a hallmark of Hubbard model physics at strong coupling ($U \gg t$), Mott insulating states have been observed in various TMD bilayers such as WSe$_2$/WS$_2$ at the filling of $n = 1$ electron (or hole) per moiré unit cell [6,7]. The existence of local magnetic moments in the Mott state is directly revealed by magnetic circular dichroism (MCD) [7,9,11]. At low magnetic fields, the temperature dependence of the MCD signal shows Curie-Weiss behavior with a negative Weiss constant that indicates antiferromagnetic exchange interaction $J = 4t^2/U > 0$. At low temperature, the MCD signal saturates at a field $B^*$ where the Zeeman energy $g\mu_B B^*$ becomes comparable to $J$.

Up to now, theoretical studies of TMD moiré materials based on Hubbard model have mainly focused on the insulating states [13–20]. Much less explored are the metallic states that appear ubiquitously at generic filling fractions. Kinetic magnetism in the metallic states is beginning to be studied [1,21,22]. Evidence of heavy Fermi liquids is being observed in MoTe$_2$/WSe$_2$ [8,23,24]. An intriguing open question is whether moiré Hubbard systems host metallic states that are fundamentally distinct from Fermi liquids.

In this work, we predict a pseudogap metal phase in doped moiré Mott insulator at $n < 1$ on the triangular lattice. This phase is realized under a certain range of magnetic (Zeeman) fields and evidenced by an intermediate magnetization plateau corresponding to a total spin $S$ that is set by the doping density $\delta = 1 - n > 0$:

$$S_p = N(1 - 3\delta)/2, \tag{1}$$

where $N$ is the number of unit cells. Our pseudogap metal is a non-Fermi liquid distinct from an ordinary metal by the presence of an energy gap to adding/removing an electron. Therefore, photoemission and tunneling measurements will find a single-particle gap at the Fermi level despite that the state is conducting and compressible. The pseudogap metal contrasts sharply with the fully spin polarized state at higher fields which has total spin $S_m = N(1 - \delta)/2$ and is a conventional Fermi liquid, as well as the zero-field state which is a metallic antiferromagnet with 120° order.

The microscopic origin of the pseudogap metal at $n = 1 - \delta$ can be traced to the nature of charge $-e$ quasiparticle in the Mott insulator at $n = 1$. For the simplicity of notation, we consider electron (with charge $e$) filling of the moiré conduction band, so that $n < 1$ corresponds to hole doping of the moiré Mott insulator. As we showed in recent work with Davydova [1], for the triangular lattice Hubbard model in the strong-coupling regime $U \gg t$, in a wide range of magnetic fields, the undoped Mott insulator is fully spin polarized, but the ground state with one doped hole is not, but contains one spin-flip that is bound to the hole [25], resulting in an itinerant spin polaron that lowers the total energy. Remarkably,

the formation of spin polaron has a kinetic origin associated with the correlated motion of the hole and spin-flip. The binding energy between the hole and spin-flip depends on the center of mass momentum $\boldsymbol{k}$ and reaches the maximum at $\boldsymbol{k} = 0$, which is on the order of the hopping amplitude $t$. In the limit $U \to \infty$, we found by exact solution $\epsilon_b(\boldsymbol{k} = 0) \approx 0.42t$ (also obtained in Ref. [25]). Therefore, for intermediate magnetic fields $h_1 < h < h_2$ with $h_1 \sim J$ and $h_2 \sim t$, the low-energy charge $-e$ quasiparticle of the spin-polarized Mott insulator is the spin polaron instead of the bare hole.

Importantly, while bare hole has spin $s = \frac{1}{2}$ relative to the undoped and fully polarized Mott insulator, spin polaron has $s = \frac{3}{2}$ due to the extra spin-flip it carries. Due to the difference in their spin quantum numbers, these two types of quasiparticles can be experimentally distinguished by measuring the lower Hubbard band edge as a function of the magnetic field [1]. Following our theoretical prediction, a recent compressibility measurement reported evidence of a transition from spin polaron to bare hole excitation as $h$ increases [26].

This work is concerned with various types of metallic states at finite hole density ($n = 1 - \delta$) that arise under the application of a magnetic field. Of particular interest is that for small hole doping and at intermediate magnetic fields, a dilute gas of itinerant spin polarons may be formed, leading to an unconventional metal [1]. From the fact that a spin polaron carries spin $s = \frac{3}{2}$, it immediately follows that the total spin of the spin polaron metal $S_p$ is *locked* to the doping level as given by Eq. (1), thus leading to a magnetization plateau with incomplete spin polarization. It also follows that there is an energy gap to adding or removing spin-$\frac{1}{2}$ electrons, which are high-energy excitations orthogonal to the underlying spin-$\frac{3}{2}$ spin polarons. Thus, our spin polaron metal is a pseudogap metal distinct from Fermi liquids. This picture is supported by our theoretical and numerical studies to be presented below.

We study the triangular lattice Hubbard model for TMD heterobilayers at electron filling $n \leq 1$ ($n$ is the number of electrons per unit cell):

$$ H = -t \sum_{\langle i,j \rangle} \left( c_{i\sigma}^{\dagger} c_{j\sigma} + h.c. \right) + U \sum_i n_{i\uparrow} n_{i\downarrow} + \frac{h(N_{\uparrow} - N_{\downarrow})}{2} \,, $$

where $c_{i\sigma}^{\dagger}(c_{i\sigma})$ is the fermion creation (annihilation) operator for spin $\sigma$ on site $i$, $n_{i\sigma} = c_{i\sigma}^{\dagger} c_{i\sigma}$ is the number operator and $h$ is the external magnetic field that couples to the $z$-component of the total spin $S_z$. We consider the case of large moiré wavelength, where the kinetic energy $t$ is much smaller than the onsite repulsion $U$.

At half-filling ($n = 1$), the ground state is an antiferromagnetic Mott insulator whose magnetic property is governed by the Heisenberg model on the triangular lattice: $H_J = J \sum_{\langle ij \rangle} \boldsymbol{s}_i \cdot \boldsymbol{s}_j$. In the experimentally relevant strong-coupling regime $U \gg t$, the exchange interaction $J$ is very small and the Mott insulator becomes fully polarized above a small magnetic field $h_1 = \frac{9}{2} J$, which is on the order of 1 T for $WSe_2/WS_2$ [7].

In contrast, at finite doping, magnetism arises predominantly from the kinetic motion of doped charges, with an energy scale set by the hopping amplitude $t$ that is much larger than $J = 4t^2/U$. To highlight the kinetic mechanism for magnetism, we shall mainly focus on the Hubbard model in the infinite-$U$ limit.

We use exact diagonalization and density matrix renormalization group (DMRG) methods [27] to study the ground state of $H$ as a function of magnetic field and doping. ED calculation is performed on two-leg ladders ($L_y = 2$) with $L_x$ between 8 and 21. To reduce the finite-size effects, periodic and anti-periodic boundary conditions in $x$ direction are used for even and odd number of holes, respectively. Since particle number and total spin $S_z$ are conserved, we divide the full Hilbert space into ($N_{\uparrow}, N_{\downarrow}$) sectors to reduce the computational cost.

We further perform quantum number conserving density renormalization group (DMRG) calculation [27–29], as implemented in the ITensor package [30], for two-leg ladders and

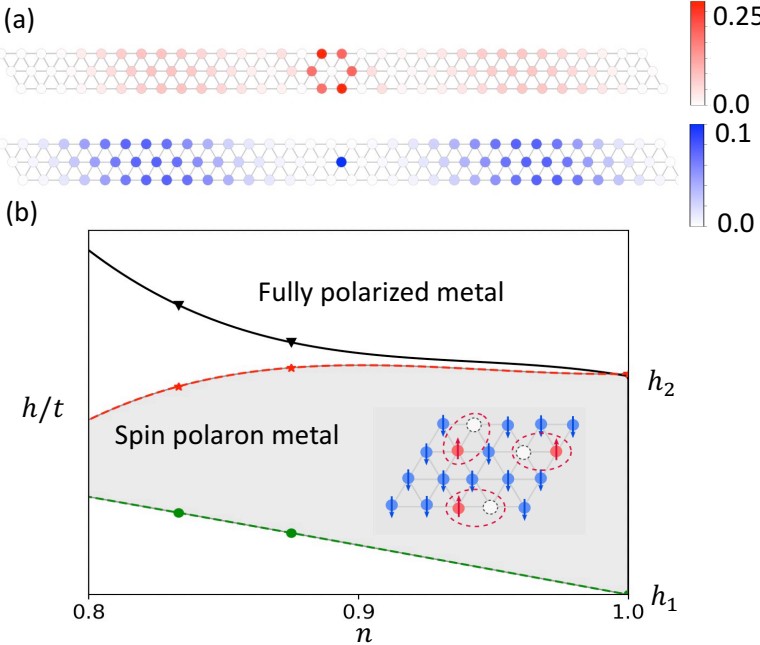

Figure 2: (a) Hole-spin correlation function $C_{hs}(i,o)/\delta$ in a $33 \times 3$ cylinder with 3 spin polarons, where holes are tightly bounded to the spin flips. The bottom panel is the Hole-Hole correlation function $C_{hh}(i,o)/\delta$. $o$ is the center of cylinder. (b) Schematic figure of doping- and field-dependent phase diagram. The data points are taken from DMRG on a $24 \times 3$ cylinder. Here N is the number of sites and Nh is the number of holes.

multi-leg cylinders ($L_y = 3, 4$ and $6$) with open boundary condition along the $x$ direction, reaching system sizes up to $40 \times 2$, $32 \times 3$ and $12 \times 6$. The convergence of our DMRG calculation is improved by keeping track of the basis transformations and using them to construct a good initial guess for the next step wavefunctions. We introduce a random noise of $10^{-6}$ to $10^{-8}$ at first few steps to avoid the local minimum trapping. The maximum bond dimension and cutoff is set to be 50000 and $10^{-7}$, the convergence criteria is $10^{-7}$ of the total energy. As a benchmark, we compare the ground state property of two-leg ladders with ED and find excellent agreement.

Fig. 1 shows the ground state magnetization (total spin $S$) at various hole doping densities as a function of magnetic field $h$. For the two-leg ladder at $\frac{1}{8}$ and $\frac{1}{6}$ dopings, magnetization curves $S(h)$ clearly converge to the thermodynamic limit with increasing system sizes $L_x$. Full spin polarization is attained at high fields $h > h_2 \sim t$, showing that kinetic energy of holes governs magnetic properties in hole-doped Mott insulator at large $U/t$. As the magnetic field is reduced, a magnetization plateau is observed over an intermediate range of fields, with the corresponding total spin $S_p$ equal to Eq.(1). This is exactly expected from the spin polaron picture: every doped hole is bound to a single spin-flip, so that the number of spin flips is equal to the number of doped holes.

As shown in Fig.1(c), magnetization plateau is also found in three-leg cylinders for various hole dopings up to at least $\delta = \frac{1}{6}$. For $\delta = \frac{1}{12}$, the magnetization curves at two different system sizes $L_x = 24, 36$ nearly coincide, showing the convergence to the thermodynamic limit. As in the two-leg case, the spin polarization on the magnetization plateau $S_p$ as a function of doping exactly matches the formula Eq.(1) expected for the spin polaron metal. With increasing doping, the width of magnetization plateaus shrinks as interaction effect between spin polarons becomes important.

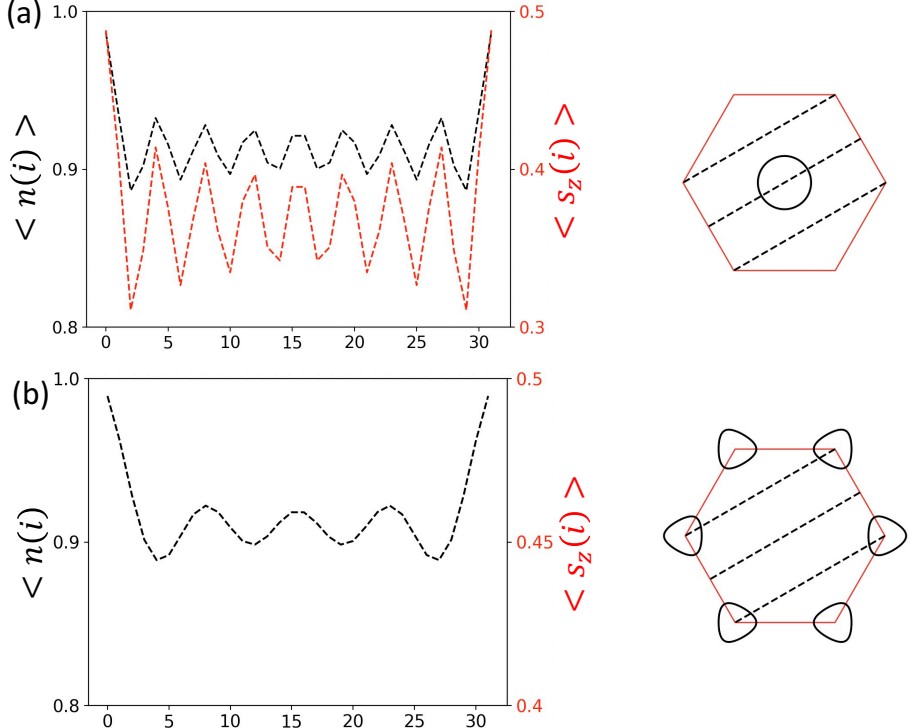

Figure 3: (a) Friedel oscillation of charge and spin densities in $32 \times 3$ cylinder with $\delta = \frac{1}{12}$ hole doping in spin polaron metal and $\delta$ is the hole density.; (b) Friedel oscillation in fully polarized metal of the same system. The two metallic states at the same density have different Fermi wavevectors, leading to different oscillation periods.

The presence of spin polaron is further supported by the spatial correlation between hole and spin-flip in the ground state. We calculate the real-space correlation function between hole and minority spin (spin-flip): $C_{hs}(i, j) = \langle n_h(i)n_\downarrow(j) \rangle$, where $n_h(i) = 1 - \sum_\sigma c_{i\sigma}^\dagger c_{i\sigma}$. As shown in Fig. 2 for three-leg cylinder, in the presence of a spin-flip, there is a very high probability ($> 72\%$) of finding a hole on its nearest-neighbor sites, indicating a tightly bound state of hole and spin-flip. We also calculate the density correlation $C_{hh}(i, j) = \langle n_h(i)n_h(j) \rangle$ and find doped holes stay away from each other, indicating the repulsive interaction between spin polarons. These correlation functions also show that at small doping, each hole spreads over many sites, consistent with the itinerant character of spin polarons.

Our recent theoretical study [1] shows that the energy dispersion of a single spin polaron has a unique minimum at $\Gamma$ with an effective mass significantly larger than the mass of bare hole. Then, at small hole doping, the dilute gas of spin polarons has a small Fermi surface centered at $\Gamma$. In contrast, the fully polarized state at high fields is a dilute gas of bare holes, which has two disconnected Fermi pockets around $K$ and $K'$ valley. Therefore, for a given doping density, the spin polaron metal and fully polarized metal, which appear at different ranges of magnetic field, have distinct Fermi surfaces with different Fermi wavevectors.

We now demonstrate the existence of Fermi surface at small doping by observing Friedel oscillations of charge and spin densities in the presence of boundaries. The period of Friedel oscillation is given by $2\pi/2k_F$, where $k_F$ is the Fermi wavevector. Fig.(3) shows density distributions in real space at $\delta = \frac{1}{12}$ hole doping for $32 \times 3$ cylinder with open boundary condition in the $x$ direction. Pronounced Friedel oscillations are observed in both spin polaron metal and fully polarized metal. The periodicity of the oscillation is $4a$ in spin polaron metal, and $8a$ in fully polarized metal.

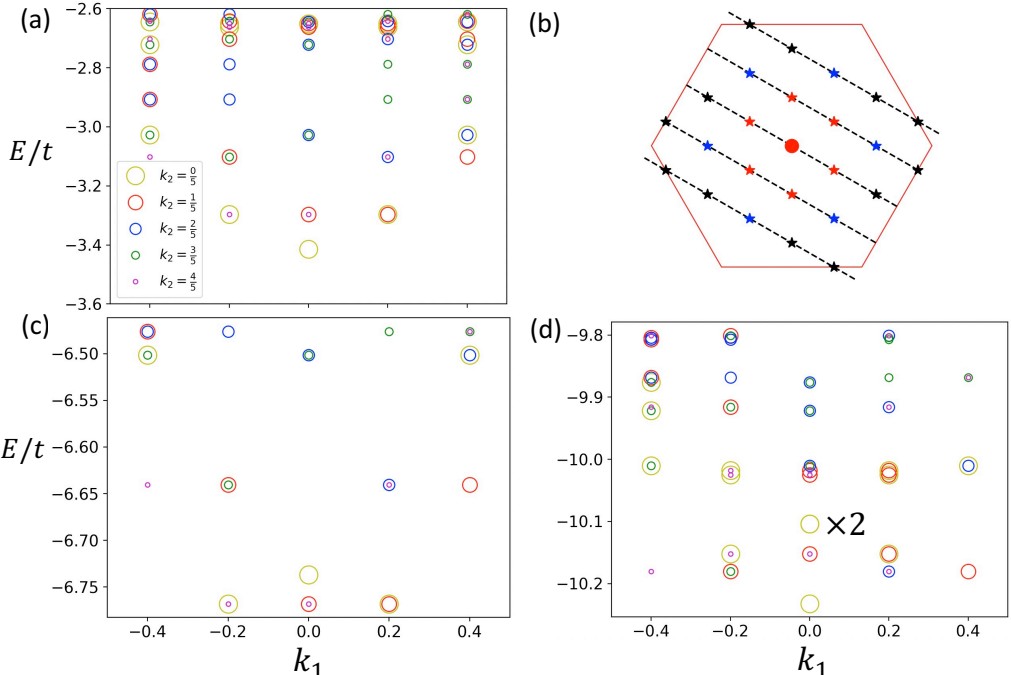

Figure 4: (a) Single spin polaron energy dispersion and (b) momentum mesh for 5×5 lattice, the ground state and six excited states are marked as large and small red dots. Energy levels for (c) two filled spin polarons with six-fold degenerate ground states, and (d) three filled spin polarons, with (1+2) fold at Γ and two six-fold degenerate energy levels at nearby momemta.

These results can be understood straightforwardly by taking into account of the finite circumference of the cylinder. In this geometry, the 2D Fermi sea is sliced into a number of 1D Fermi sea at various $k_y = 2\pi n/L_y$ with $n = 0, ..., L_y-1$. For $L_y = 3$ and $\delta = \frac{1}{12}$, in spin polaron metal, the line $k_y = 0$ cuts through a single Fermi surface centered at Γ. The corresponding $2k_F$ leads to an oscillation period $a/(\delta L_y) = 4a$. In contrast, in fully polarized metal, the Fermi wavevector is halved due to the presence of two disconnected Fermi surfaces, leading to a doubled period of $8a$.

In the presence of single spin polaron, we calculate the momentum dependent energy levels at the sub-Hilbert space with single hole and spin flip using ED. A 2D geometry with system size $(L_x, L_y) = (5,5)$ is considered here. The corresponding single spin polaron dispersion has an unique minimum at Γ and six degenerate excited states surrounding Γ as shown in Fig. 4(a,b). Due to the weak interacting nature, spin polarons are filled to the system from the lowest energy levels. When two spin polarons are filled, we get six degenerate ground state energy levels in Fig. 4(c) from the superposition of Γ and six-fold degenerate excited states in single spin polaron Hilbert space. In the case of three spin polarons, we find the lowest energy sectors are directly constructed from the superposition of Γ state and two of six second lowest energy states, leading to $C_6^2 = 15$ many-body states. For the 15 lowest energy levels in 4(d), we have (1+2) at Γ and (6+6) at nearby momenta, well consistent with the filling picture of spin polarons.

Our results so far have identified an unconventional metallic phase—a dilute gas of spin polarons—in the triangular lattice Hubbard model at small hole doping and intermediate magnetic field. A hallmark of spin polaron metal is that its zero-temperature magnetization is determined solely by the doping level as given by Eq.1. It is a compressible state with a single Fermi surface centered around Γ.

We now demonstrate another defining feature of spin polaron metal: it has a single-particle gap, i.e., adding/removing an electron costs finite energy. Intuitively, the reason is obvious: the constituent particles forming the metallic state are spin polarons carrying spin-$\frac{3}{2}$, while the electron carrying spin-$\frac{1}{2}$ is a high-energy excitation. To show the single-particle gap explicitly, let us denote the ground state energy at a given total particle number $N$ and total spin $S$ by $E(N,S) = E_0(N,S) - hS$, where $E_0$ is the ground state energy at $h = 0$. Minimizing $E(N,S)$ over all possible values of $S = 0,...,N/2$ yields the $N$-particle ground state energy $E_N \equiv \min_S E(N,S)$ as well as the corresponding spin which is denoted as $S_N$. Adding an electron necessarily increases or decreases the total spin by $\frac{1}{2}$. The single-particle gap is thus defined by

$$\Delta E_{e,s} = E(N+1, S_N + s) - E_N - \mu, \qquad (2)$$

with $s = \pm \frac{1}{2}$ depending on the added electron being spin $\uparrow$ or $\downarrow$. $\mu$ is the chemical potential defined by $\mu = \frac{\partial E_N}{\partial N}$. For metallic states in the thermodynamic limit $N \to \infty$, $\mu = E_{N+1} - E_N$. Then, the single-particle gap is equal to $\Delta E_{e,s} = E(N, S_{N-1} + s) - E_N$. In the spin polaron metal phase, we have $S_{N-1} = S_N - s_0$ with $s_0 = \frac{3}{2}$ because the $(N-1)$-particle state has one more spin polaron than the $N$-particle state. It thus follows that the single-particle gap is equal to the spin gap, i.e., the energy cost resulting from the inevitable spin mismatch $s - \frac{3}{2} \neq 0$ between spin polaron and the added electron:

$$\Delta E_{e,s} = E(N, S_N + s - s_0) - E_N. \qquad (3)$$

Importantly, the presence of magnetization plateau over a finite range of magnetic fields implies the existence of spin gap in the spin polaron metal (otherwise the total spin would change continuously with $h$). We thus conclude that spin polaron metal has a single particle gap. It should be contrasted with the fully polarized metal at high field, which also has a spin gap. In that case, $s_0 = \frac{1}{2}$ hence $\Delta E_{e,+\frac{1}{2}} = 0$, i.e., there is no gap to adding an electron of spin $\uparrow$.

To summarize, the main finding of our work is spin polaron metal in doped Mott insulator on the triangular lattice under a magnetic field. It is a remarkable state of matter at generic fillings, which is conducting and compressible similar to an ordinary metal, but has a spin gap and a single-particle gap. Its existence can be can experimentally established by a zero-temperature magnetization $S_p$ that depends only on the doping as given by Eq.(1). Increasing the magnetic field drives a transition from the spin polaron metal to the fully polarized metal, accompanied by a change of Fermi surface volume that can be detected by the change of Landau level degeneracy and quantum oscillation frequency.

Finally, we briefly discuss the ground state of infinite-$U$ Hubbard model at smaller magnetic fields $h < h_1$. As $h$ is reduced, the magnetization decreases continuously to zero at $h = 0$. For three-leg cylinders, we do not observe any additional plateau that could be associated with bound state between hole and multiple spin-flips [25, 31]. At zero field, we find that small doping induces strong antiferromagnetic correlation wavevector $K, K'$ consistent with 120° order, consistent with a recent study [32]. In the magnetization curves shown in Fig. 1, the magnetization plateau stands out as the most prominent feature of doped Mott insulator on the triangular lattice, which we identify as the hallmark of a pseudogap metal composed of itinerant spin polarons.

## Acknowledgments

We thank Margarita Davydova for collaboration on a recent work [1] which led to this study.

# A  Pseudogap metal and magnetization plateau from doping moiré Mott insulator

## A.1  Quantum number conserving Density Matrix Renormalization Group

In this section, we present the detailed description of our DMRG calculation. We employ the DMRG algorithm as implemented in ITensor package [30] on the finite ladder and cylinder. And the $x$ direction is chosen as the long direction with open boundary condition to reduce entanglement and bond dimension. We utilize $2 \times L_x$ ($L_x$ up to 40) ladder and $3 \times L_x$ ($L_x$ up to 32) cylinder and calculation hole doping up to $\frac{1}{6}$ around half-filling. The site index is chosen to increase along the narrow $y$ direction. For small system size (site number less than 40), a random matrix product state (MPS) with fixed number of holes $N = N_\uparrow + N_\downarrow$ and spin flips $N_\downarrow$ is generated as the initial wavefunction. Without spin coupling, the $tJ$ and full Hubbard Hamiltonian have $SU(2)$ symmetry, and the DMRG calculation preserves the quantum numbers during the convergence steps. Consequently, we parallelize the magnetic plateau calculations over different particle number and spin sectors.

When the initial MPS is close to the global minimum, robust convergence can be achieved with a few DMRG steps and moderate bond dimensions. However, if the initial MPS is far from the global minumum then there is no guarantee that DMRG will be able to find the true ground state. In our problem, occupation and spin quantum number are conserved to reduce the Hilbert space dimension. The convergence problem is much more significant since the search space is more constrained.

To improve the numerical stability, here we add noise term from $10^{-6}$ to $10^{-10}$ to avoid local mimimum trapping effect. More importantly, we create the initial MPS with properties close to the ground state in terms of charge configuration and spin ordering. Especially for the case of multi spin polarons at two or three-leg ladders, we can approximate the trial MPS of large system from the combination of converged MPS of a small system. Therefore, the electron density is spread out over the whole system and the spin flips are binded to the hole, as expected for the final MPS. The creation of initial MPS greatly reduces the computation cost and bond dimension for large system size, and allows the computation for 16 holes in a $3 \times 32$ cylinder, which has a Hilbert space dimension $1.7 \ast 10^{34}$ at the sector with $N_\downarrow = 16$ in the $tJ$ basis.

For the accurate evaluation of magnetic plateau and correlation function, we set energy convergence criteria as $10^{-7}t$, the cutoff as $10^{-8}t$, and maximum bond dimension as 40000. We will compare the energy with exact diagonalization for the case with small Hilbert space in the next section and the energy difference is normally within $10^{-8}t$. With the converged ground state MPS, we calculate the electron density distribution, hole-hole $\langle n(0)n(x) \rangle$ and hole-spin $\langle n(0)s_z(x) \rangle$ correlations in real space, where we defined $n(i) = 1 - \sum_\sigma c_{i\sigma}^\dagger c_{i\sigma}$ as the hole density operator.

**Comparison with exact diagonalization.**  We compare the ground state energy from DMRG with exact diagonalization. At a computing node with $500GB$ memory, the up limit for Hilbert space dimension in exact diagonalization is $2 \times 10^9$ if we use float128 to represent the basis and Hamiltonian. Within the $tJ$ model at $J = 0$ limit, we compare the ground state energy, real space electron and spin density for 4 holes at $2 \times 12$ ladder. As shown in Fig. 5, the magnetization plateaus and real space electron density ($N_\downarrow = 4$) plots from ED and DMRG almost overlap with each other.

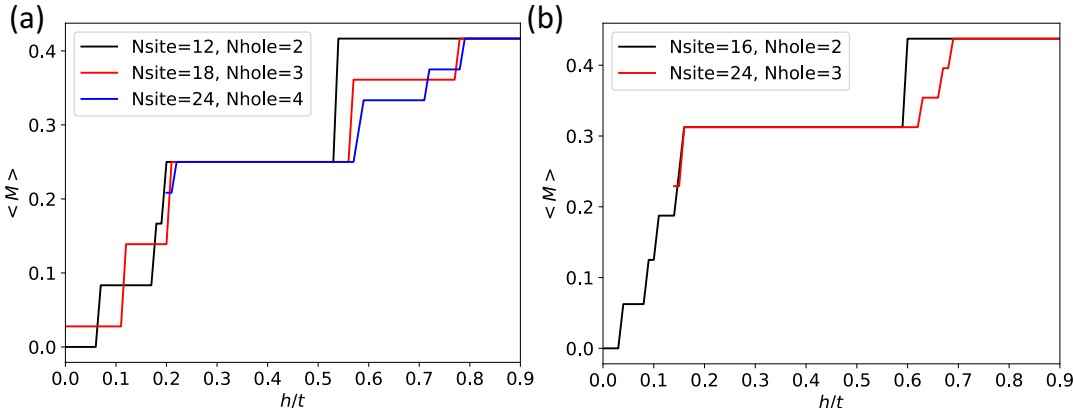

Figure 5: The magnetic moment $M$ under magnetic field obtained by exact diagonalization at infinite U limit and 1/6 and 1/8 hole doping on two-leg ladder with $L_x = 6, 9, 12$ and periodic boundary condition in $x$ direction.

## A.2 Filling rule of spin polarons

In the presence of single spin polaron, we calculate the momentum dependent energy dispersion at the spin polaron sector from ED. For system size $(L_x, L_y) = (6, 3)$, the fully polarized state locates at $M = (1/2, 0)$ (applies to even $L_x$ in general), and we subtract the background momentum in the spin polaron dispersion. The ground state momentum is at $\Gamma$, followed by the two fold degeneracy near $\Gamma$ and closely spaced energies from other momentums.

Due to the weak interacting nature of spin polarons, it is expected that spin polarons are filled to the system from the lowest energy levels similar as the weakly-interacting particles. For $(L_x, L_y) = (6, 3)$ with periodic boundary condition, we find the filling rules works up to three spin polarons. By applying the anti-periodic boundary condition at $x$ direction, the Fermi surface of single spin polaron is shifted by half momentum spacing $\frac{1}{2L_x}G_x$. And we find the filling rule is valid up to 4 spin polarons, with the energy degeneracy and multi-spin polaron ground state momenta determined by single spin polaron dispersion.

We then apply the anti-periodic boundary condition to split the six-fold degeneracy. With anti-BC at $x$ direction, the spin polaron dispersion has two degenerate minima at $G$ and $(-\frac{1}{5}, 0)$, followed by two degenerate energies at $(-\frac{1}{5}, \frac{4}{5},)$ and $(0, \frac{1}{5})$. The wavefunction of two and three spin polarons are constructed from the superposition of four states from energy ordering. For the case of anti-BC at $x$ and $y$ direction, the filling rules also apply.

## A.3 Antiferromagetism at finite doping

We study the ground-state properties of infinite U Hubbard model at total spin $S^z = 0$ spin sector [32]. At filling factor $n = 1$, the spin exchange $J = 4\frac{t^2}{U}$ is vanishing small at infinite U limit. At finite hole doping side, we now consider antiferromagnetic interaction at zero magnetic field. In the ultra-strong coupling $U \gg t$ limit, spin correlation arises from kinetic term. And we probe the ground state magnetic property of the $J = 0$ $tJ$ model by calculating the spin structure factor $S_{\mathbf{q}}(\mathbf{Q}) = \frac{1}{N}\sum_{i,j}\left\langle S_i^z S_j^z\right\rangle e^{i\mathbf{Q}\cdot(\mathbf{r_i}-\mathbf{r_j})}$. Here we consider the number of holes from 0 to 16 in a $3 \times 24$ and $4 \times 18$ cylinder, covering 0% to 22% doping density. At zero magnetic field, the total $\left\langle S_i^z\right\rangle$ is vanishingly small. In the DMRG calculation, we target at the spin sector with $\left\langle S_i^z\right\rangle = 0$ by enforcing $N_\uparrow = N_\downarrow$.

**Doping dependence of kinetic antiferromagnetic exchange** And we confirmed that $S_{\mathbf{q}}(\mathbf{Q})$ is zero at all momentum. By doping even number of holes into the $24 \times 3$ ladder, we find the $S_{\mathbf{q}}(\mathbf{Q})$ becomes nonzero and sharply peaks at $\mathbf{Q} = \mathbf{K}$ momentum as shown in the contour

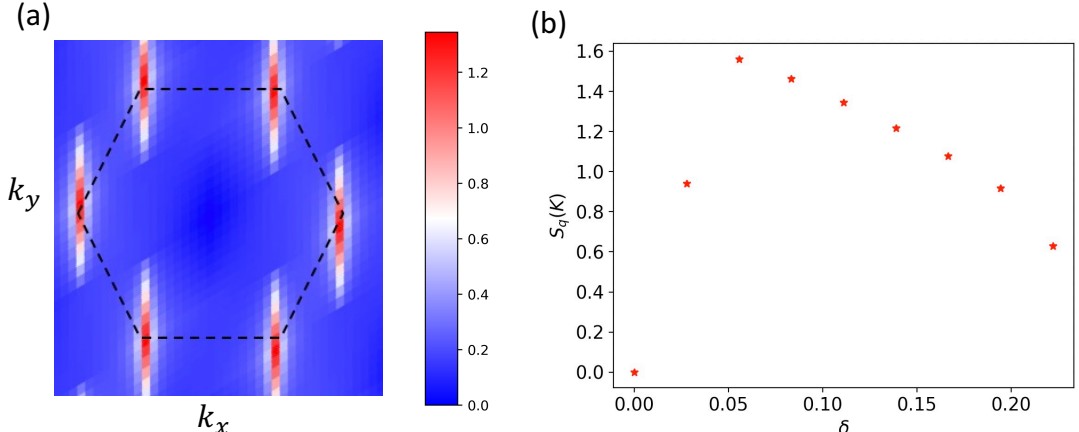

Figure 6: The static spin structure factor $S_q(Q)$ on $24 \times 3$ cylinder for infinite $U$ Hubbard model. (a) Contour plot of $S_q(Q)$ at hole doping $\delta = 5.5\%$, which shows a peak at momentum $Q = K$. (b) Maximum $S_q(Q)$ at momentum $K$ as a function of hole doping, indicating an enhanced antiferromagnetic exchange at finite doping density.

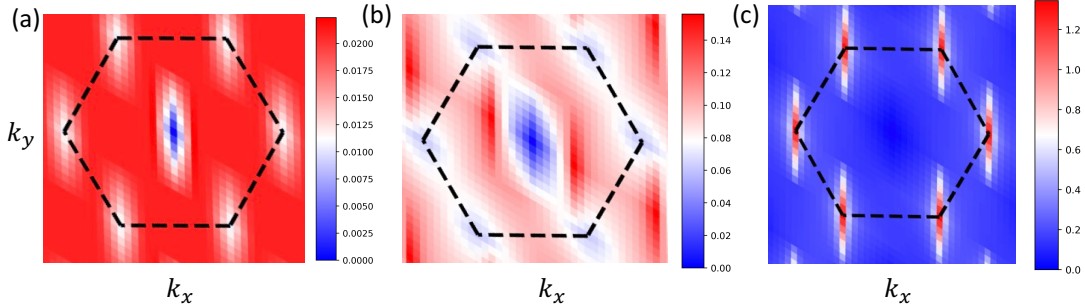

Figure 7: The static spin structure factor $S_q(Q)$ on $24 \times 3$ cylinder for infinite $U$ Hubbard model with hole doping $\delta = 5.5\%$ (6 holes). (a) Fully polarized spin sector. (b) Spin polaron sector with 6 spin flips. (c) Antiferromagnetic spin sector with total $< S_z >= 0$.

plot in 6(c), indicating the kinetic induced antiferromagnetism with 120° noncollinear order. In 6(b), we plot the maximum $S_q($K as a function of hole doping density. Starting from massive degenerate spin disorder phase at $n = 1$, the antiferromagnetism gets greatly enhanced at light doping region with a peak at $\delta = 5.5\%$, and suppressed with further enlarged hole doping. We further calculate the spin exchange as a function of total magnetization $< S_z >$ in a $24 \times 3$ cylinder. For the fully polarized spin sector, the spin structure factor map has deeps at $\Gamma$ and $K$ momentum. When the number of spin flip is equal to the number of holes, the spin polaron spin structure map has two sharp peaks around $\Gamma$ momentum as shown in Fig. 7.

## A.4 Magnetization plateau in the finite-$U$ Hubbard model

As described in the main text, the width of magnetization plateau stay unchanged when $U$ is reduced from infinite limit. We now present the detailed calculation of $U = 10, 20, 40$ Hubbard model. In real material systems, the onsite repulsion $U$ is always finite. We consider the case of finite $U$, and the corresponding magnetization plateau within the Hubbard model.

To properly include the effect of double occupancy at finite coupling strength, we directly diagonalize the Hubbard model with double occupancy allowed Hilbert space for U=10, 20, 40

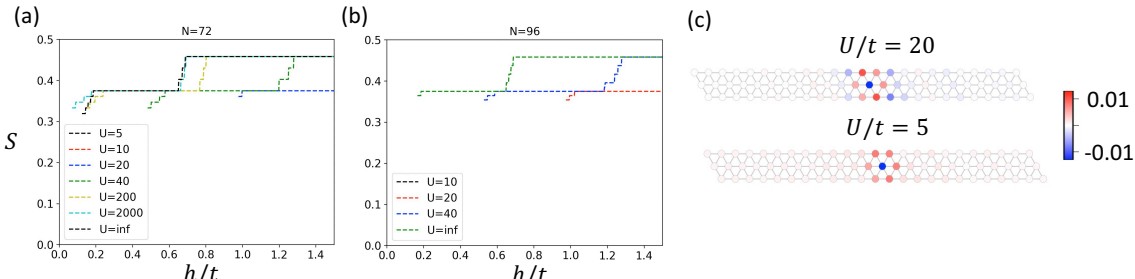

Figure 8: The total spin $S$ under magnetic field obtained by DMRG for different $U$ at (a) $\delta = 1/12$ for $24 \times 3$ cylinder, (b) $\delta = 1/12$ for $24 \times 3$ cylinder. (c) Hole-minority spin correlation function normalized by hole density $C_{hs}(i, o)/\delta$ for 6 spin polarons in a $25 \times 3$ cylinder. Upper and bottom panel for $U/t = 5, 20$, respectively.

using DMRG. For three-leg ladder with $L_x = 24$ with finite $U$, the magnetization plateaus get shifted to higher magnetic field, which is consistent with enlarged saturation field [1]. The width of magnetization plateau stay unchanged with decreasing onsite interaction down to $U = 10t$. Therefore, we conclude the presence of spin polaron metal at wide range of onsite repulsion.

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
