# Peer review of "Pseudogap metal and magnetization plateau from doping moiré Mott insulator"

_SciPost Physics Core, doi:SciPost Phys. Core 6, 038 (2023)_

## Round 1 · Referee Report · Anonymous (Referee 1) · 2023-3-12

Report

Following their original study [ref. 1], the authors have calculated the magnetic response of the spin polaron phase in a hole-doped triangular Mott insulator. They found magnetization plateaus at intermediate magnetic fields (in between J and t) as a unique signature of the spin polaron phase. They have further calculated the Fermi surface and the energy dispersion of the spin polaron metal, and pointed out the presence of a single particle or spin gap in this novel metallic state, i.e. a pseudo-gap metal.

The study is timely and highly relevant to ongoing experimental studies of the triangular Hubbard physics in semiconductor moiré materials. Compared to their original study [ref. 1], the authors have predicted specific experimental observables that can be immediately tested by experiments. In this sense, the study is important and suitable for publication in SciPost. I only have a few minor comments that I would like the authors to address.

  1. The labels (N and N_h) in figure 1c are not defined in the main text or in the captions. I would recommend to label the curves in the doping levels delta.

  2. I recommend the authors to specify the value of delta in the caption of figure 2a.

  3. Can the authors comment on the Hall coefficient and its connection to the change in Fermi surface when the spin polaron metal is changed to a spin polarized metal with magnetic field? The authors have mentioned quantum oscillations as a probe of the Fermiology but the Hall effect could be a more realistic experimental probe in the near future.

---

## Editorial Decision

published